# Predicting Intraoperative Hypothermia Burden during Non-Cardiac Surgery: A Retrospective Study Comparing Regression to Six Machine Learning Algorithms

**DOI:** 10.3390/jcm12134434

**Published:** 2023-06-30

**Authors:** Christoph Dibiasi, Asan Agibetov, Lorenz Kapral, Sebastian Zeiner, Oliver Kimberger

**Affiliations:** 1Department of Anaesthesia, Intensive Care Medicine and Pain Medicine, Medical University of Vienna, Währinger Gürtel 18-20, 1090 Vienna, Austria; christoph.dibiasi@meduniwien.ac.at (C.D.); sebastian.zeiner@meduniwien.ac.at (S.Z.); 2Ludwig Boltzmann Institute Digital Health and Patient Safety, Währinger Straße 104/10, 1180 Vienna, Austria; lorenz.kapral@dhps.lbg.ac.at; 3Center for Medical Statistics, Informatics and Intelligent Systems, Institute of Artificial Intelligence, Medical University of Vienna, Währinger Straße 25a, 1090 Vienna, Austria

**Keywords:** anesthesia, surgery, hypothermia, prediction, machine learning

## Abstract

Background: Inadvertent intraoperative hypothermia is a common complication that affects patient comfort and morbidity. As the development of hypothermia is a complex phenomenon, predicting it using machine learning (ML) algorithms may be superior to logistic regression. Methods: We performed a single-center retrospective study and assembled a feature set comprised of 71 variables. The primary outcome was hypothermia burden, defined as the area under the intraoperative temperature curve below 37 °C over time. We built seven prediction models (logistic regression, extreme gradient boosting (XGBoost), random forest (RF), multi-layer perceptron neural network (MLP), linear discriminant analysis (LDA), k-nearest neighbor (KNN), and Gaussian naïve Bayes (GNB)) to predict whether patients would not develop hypothermia or would develop mild, moderate, or severe hypothermia. For each model, we assessed discrimination (F1 score, area under the receiver operating curve, precision, recall) and calibration (calibration-in-the-large, calibration intercept, calibration slope). Results: We included data from 87,116 anesthesia cases. Predicting the hypothermia burden group using logistic regression yielded a weighted F1 score of 0.397. Ranked from highest to lowest weighted F1 score, the ML algorithms performed as follows: XGBoost (0.44), RF (0.418), LDA (0.406), LDA (0.4), KNN (0.362), and GNB (0.32). Conclusions: ML is suitable for predicting intraoperative hypothermia and could be applied in clinical practice.

## 1. Introduction

Inadvertent intraoperative hypothermia is a common but often preventable complication during surgery [1]. In addition to greatly affecting patient comfort, hypothermia is thought to be associated with increased perioperative morbidity [2,3]. It also imposes a significant cost on the healthcare system [4]. Potentially underlying mechanisms of the mid- to long-term effects of intraoperative hypothermia include an increased rate of wound infections [5], impaired blood coagulation, and increased use of blood products [6]. Thermal care interventions, such as active warming prior to arrival in the operating room [7], intraoperative active warming [8], and fluid warming [9], have been shown to be effective in alleviating hypothermia and its associated complications [10] and are recommended by most guidelines [11]. Yet, at least mild hypothermia still occurs in the majority of surgical cases, even in fully equipped healthcare systems [12]. However, a recent multicenter trial demonstrated that intraoperative core temperatures as low as 35.5 °C are not associated with adverse clinical outcomes [13].

Estimating the risk of inadvertent hypothermia could enable the targeted use of thermal care interventions and thereby optimize the use of these potentially limited resources. We recently developed several regression models to predict intraoperative hypothermia of less than 36 °C and less than 35.5 °C with sufficient accuracy [14]. However, in this study, hypothermia was defined as point incidence of body temperature below two fixed and arbitrary thresholds. Hypothermia burden, defined by us as the area below the body temperature under 37 °C curve over time, may be a more appropriate outcome parameter. In addition, various studies in recent years have indicated that machine learning (ML) could be advantageous over logistic regression [15,16,17,18]. We therefore hypothesized that ML might be superior to logistic regression in predicting intraoperative hypothermia. Currently, there is no default approach to ML modelling, and each algorithm has unique strengths and weaknesses [18]. For instance, ensemble models—which integrate multiple algorithms—such as random forest (RF) and extreme gradient boosting (XGBoost) are robust to missing values but are weak at analyzing non-linear patterns in structured datasets [19]. In contrast, generative models such as linear discriminant analysis (LDA) models and Gaussian naïve Bayes (GNB) models assume that the predictor variables do not interact with each other [20].

In this study, we aimed (1) to analyze whether ML algorithms would be suitable to predict intraoperative hypothermia and (2) to compare their predictive performance among themselves and to logistic regression. We therefore trained a logistic regression model and six supervised classification ML models and assessed discrimination and calibration.

## 2. Materials and Methods

### 2.1. Data Collection

We analyzed electronic anesthesia records of the General Hospital of Vienna, a tertiary academic medical center, from the period September 2013 to March 2021. Patients were eligible when general anesthesia, neuraxial anesthesia, peripheral plexus block, or nerve block was performed for a surgical intervention and when body temperature measurements were available. Patients were excluded from the analysis if extracorporeal circulation (i.e., cardiopulmonary bypass or extracorporeal membrane oxygenation) was used during surgery or when therapeutic hyperthermia (e.g., hyperthermic intraperitoneal chemotherapy) was performed. In addition, we excluded any cases with a total duration of less than 15 min and brain-dead patients scheduled for organ donation. When a patient had multiple anesthesia cases, each case was counted separately. We exported all data from the patient data management system IntelliSpace Critical Care and Anesthesia (Philips Austria GmbH, Vienna, Austria). We performed no formal sample size calculation for this study.

### 2.2. Outcome Variables

Intraoperative body temperature was recorded at 2 min intervals. After data export, we removed artifacts in body temperature measurements utilizing the algorithm published by Sun et al. [21]. We then calculated the hypothermia burden, the area below 37 °C on the intraoperative time–body temperature curve, as our primary outcome. As we had aimed to use classification ML algorithms in this study, we assigned each patient to one of four groups, namely, no hypothermia, mild hypothermia, moderate hypothermia, and severe hypothermia, corresponding to the respective quarters of patients ranked by hypothermia burden.

We considered the binary variables “hypothermia below 36 °C, 35.5 °C, or 35 °C at a single time point” as secondary outcomes. Primary and secondary outcomes were decided upon prior to data analysis.

### 2.3. Predictor Variables

The feature set was comprised of 71 variables. Three variables described patient demographics (age (years), weight (kg), sex (male/female)); 2 variables described comorbidities (American Society of Anesthesiologists score (1–6), van Walraven comorbidity score (−19–89)) [22]; 14 variables encoded type of surgery (surgical urgency (elective/emergent/emergency), surgical billing code, and intraoperative positioning (supine, prone, Trendelenburg, anti-Trendelenburg, beachchair, leg extension, Lloyd–Davis, lithotomy, side, flexed side, other, not documented; all yes/no)); and 9 variables gave information on type of anesthesia (endotracheal intubation, laryngeal mask, fiberoptic intubation, spinal anesthesia, epidural anesthesia, upper extremity peripheral nerve block, lower extremity peripheral nerve block, peripheral nerve block at unknown location (all yes/no), and room temperature at induction of anesthesia (°C)). We included 8 preoperative laboratory parameters (hemoglobin (g/dL), leucocyte count (G/L), platelets (G/L), fibrinogen (mg/dL), c-reactive protein (mg/dL), aspartate aminotransferase (U/L), alanine aminotransferase (U/L), and creatinine (mg/dL)); 7 variables for induction medication (dosage of propofol (mg), midazolam (mg), etomidate (mg), esketamine (mg), fentanyl (µg), rocuronium (mg), cisatracurium (mg)); 4 variables for maintenance of anesthesia (propofol, remifentanil, any volatile agent, nitrous oxide (all yes/no)); and 24 variables for vital signs (systolic, mean, and diastolic blood pressure (all mmHg), peripheral transcutaneous oxygen saturation (0–100), heart rate (beats per minute), and pulse frequency (beats per minute)). Using the longitudinal vital sign measurements recorded every 15 s, we extracted the initial (i.e., first available vital signs in the record), minimum, and maximum (up to the documented start of surgery) vital signs. In addition, we collected the first available vital signs after documented induction of anesthesia.

### 2.4. Imputation of Missing Values and Feature Scaling

Imputation and feature scaling were fitted on the training set as part of a classifier pipeline, and the same pipeline was used to estimate the performance on the holdout test set. We used a single iterative imputer implementation from the scikit-learn Python package. Missing values of each feature were imputed in a round-robin fashion, where at each step each feature with a missing value is modeled as a function of other features. We used the built-in imputation scheme for the tree-based algorithms XGBoost and RF. The imputation step was fitted exclusively on training data to prevent information leakage into the model. For the algorithms that are sensitive to the scale of feature values (i.e., non-tree-based algorithms), we standardized each feature by removing the mean and scaling to unit variance. Categorical variables were label encoded. In the case of decision-tree based ML algorithms, categorical variables were one-hot encoded.

### 2.5. Prediction Models and Machine Learning Algorithms

We applied logistic regression models with different regularization penalties (L1, L2, elastic net) and the following ML algorithms:XGBoost, a gradient tree-boosting algorithm comprising a multitude of decision trees.RF, another decision-tree-based ensemble learning technique where a combined prediction of a forest of multiple random decision trees is used to obtain a more accurate and stable prediction. The key difference to XGBoost is that an RF algorithm is trained using the “bagging” technique as opposed to gradient boosting.LDA, which discriminates between classes by learning the joint probability distribution of the input and target variables.GNB, a modification of LDA where the covariance matrix is a diagonal matrix, thus drastically simplifying the computation.k-nearest neighbor (KNN), a non-parametric algorithm that classifies a new data point based on the similarity to the training set.Multi-layer perceptron (MLP) neural network, with a standard feed-forward architecture with hidden layers consisting of neurons.

### 2.6. Model Tuning, Selection, and Evaluation

We randomly split the whole dataset into training and test subsets using a ratio of 70:30 and stratified them to preserve the positive/negative class ratio. We only used the test set for the final evaluation of the optimized models, and no information leaked into the training and model tuning phases. To predict the hypothermia burden group (no, mild, moderate, or severe hypothermia), we measured the discriminative performance of all models using the weighted F1 score, which incorporates two important characteristics of predictive models—precision and recall. Precision (or positive predictive value) refers to the proportion of true positives to all predicted positives (i.e., true and false positives) and—in the context of predicting hypothermia—gives the probability that a patient predicted to develop hypothermia will truly become hypothermic. Recall (or sensitivity) is given by the ratio of true positives to all positives (i.e., true positives and false negatives). It can be interpreted as the chance that a patient will develop hypothermia as detected by a given model. The F1 score is calculated as the harmonic mean of precision and recall:F1 = 2 × (Precision × Recall)/(Precision + Recall)(1)

To predict the hypothermia burden group, we used the weighted F1 score, which averages the F1 scores for each class (no, mild, moderate, or severe hypothermia) using the number of occurrences of each class as weight. We predicted the secondary outcomes (occurrence of intraoperative temperature below 35 °C, 35.5 °C, or 36 °C) using binary prediction models and used the area under the receiver operating characteristic curve (AUROC) to assess discriminative performance. We also used AUROC to report the discriminative performance of the primary outcome prediction models for each hypothermia burden category (e.g., no hypothermia vs. the rest; mild hypothermia vs. the rest, etc.).

We assessed calibration using calibration-in-the-large, calibration intercept, and slope. Calibration-in-the-large is defined as the difference in the log-odds of the mean observed value compared to the mean predicted value; a value closer to 0 indicates better calibration. Calibration intercept and slope were calculated using the Cox method, whereby the observed binary outcome is regressed to the log odds of the estimates using a general linear model [23]. The estimated regression intercept represents the overall miscalibration, where 0 indicates good calibration, while the estimated regression slope gives the direction of miscalibration, where 1 denotes perfect calibration.

We performed hyperparameter optimization for each algorithm on the training set by employing a Bayesian search space optimization technique to estimate the best parameters for each model stochastically. For each algorithm, we let the hyperparameter optimization run for 50 trials. In each trial, a hyperparameter configuration was sampled, and a 3-fold cross-validation was used to evaluate this specific configuration. Based on the mean discrimination score (F1 for the primary outcome or AUROC for the secondary outcomes) of a 3-fold cross-validation, the optimization algorithm decided where to sample the next hyperparameter configuration. We provide the search space sampling definition for the hyperparameters as Appendix A. The best configuration of hyperparameter values was chosen based on the cross-validated discrimination score (the higher the better) and was evaluated using the test set.

### 2.7. Software

Data export was performed using Microsoft SQL Server Management Studio 18 (Microsoft, Redmond, WA, USA). Data analysis was performed using R version 4.1.2 [24] and Python version 3.10.7. We used the official Python package for XGBoost [25] and the scikit-learn Python package [26] to implement the remaining ML algorithms. The Python package *skopt* [27] was used for the Bayesian hyperparameter optimization.

## 3. Results

We analyzed data from 140,241 eligible surgical cases performed between September 2013 and March 2021; 87,116 cases were included in the final analysis (Figure 1). Baseline data of included patients are given in Table 1.

The primary outcome was hypothermia burden, calculated as area under the curve of body temperature below 37 °C over the intraoperative course for each individual patient. Overall median (interquartile range (IQR)) hypothermia burden was 1.01 °C·h (0.49–1.87). We then split the whole dataset into four equally sized quarters of patients stratified by hypothermia burden. In total, 19,484 patients were in the quarter with the lowest hypothermia burden (temperature AUC range 0–0.44 °C·h; i.e., “no hypothermia”), 22,220 patients were in the second quarter (temperature AUC range 0.44–0.96 °C·h; i.e., “mild hypothermia”), 22,800 patients were in the third quarter (temperature AUC range 0.96–1.82 °C·h; i.e., “moderate hypothermia”), and 22,612 patients were in the fourth quarter with the highest hypothermia burden (temperature AUC range > 1.82 °C·h; i.e., “severe hypothermia”). The median (IQR) hypothermia burdens in the respective quarters were 0.23 °C·h (0.08–0.34), 0.68 °C·h (0.56–0.81), 1.31 °C·h (1.12–1.54), and 2.73 °C·h (2.18–3.78).

We trained one logistic regression model and six ML algorithms (XGBoost, RF, LDA, GNB, KNN, MLP) to predict the hypothermia burden group of individual patients. Logistic regression had a weighted F1 score of 0.397. Ranked by weighted F1 score, the best performing algorithm was XGBoost with an F1 score of 0.44, an 10.74% increase compared to logistic regression (Table 2).

In addition to hypothermia burden, we predicted hypothermia based on minimum body temperature at three thresholds: <35 °C, <35.5 °C, and <36 °C. In total, 5592 patients (6%) had a minimum body temperature <35 °C; 17,240 patients (20%) below 35.5 °C, and 43,379 patients (50%) below 36 °C. We obtained the following AUROC with logistic regression: 0.701 for <35 °C, 0.692 for <35.5 °C, and 0.703 for <36 °C. Regarding the ML algorithms, XGBoost had the highest AUROC across all three temperature thresholds: 0.736 for <36°C, 0.717 for <35.5 °C, and 0.715 for <35 °C (Appendix A). In general, AUROC was lower than the values obtained from predicting the hypothermia burden quarter. We obtained good calibration with models trained to predict body temperature <36 °C, but calibration worsened when temperatures of <35.5 °C and <35 °C were to be predicted (Appendix A).

We then tested the binary predictions of individual hypothermia burden groups (i.e., whether individual patients will develop no hypothermia or not, or mild hypothermia or not, and so on). For this, we calculated AUROC. For all algorithms, AUROC was higher when predicting the occurrence of no and/or severe hypothermia compared to predicting the occurrence of mild and/or moderate hypothermia (Figure 2). Detailed results for model discrimination are given in Appendix A. Most models had acceptable calibration, with GNB being a notable exception (Table 3 and Figure 3).

In addition to hypothermia burden, we predicted hypothermia based on minimum body temperature at three thresholds: <35 °C, <35.5 °C, and <36 °C. In total, 5592 patients (6%) had a minimum body temperature <35 °C; 17,240 patients (20%) below 35.5 °C, and 43,379 patients (50%) below 36 °C. We obtained the following AUROC with logistic regression: 0.701 for <35 °C, 0.692 for <35.5 °C, and 0.703 for <36 °C. Regarding the ML algorithms, XGBoost had the highest AUROC across all three temperature thresholds: 0.736 for <36 °C, 0.717 for <35.5 °C, and 0.715 for <35 °C (Appendix A). In general, AUROC was lower than the values obtained from predicting the hypothermia burden quarter. We obtained good calibration with models trained to predict body temperature < 36 °C, but calibration worsened when temperatures of <35.5 °C and <35 °C were to be predicted (Appendix A).

## 4. Discussion

In this study, we trained and evaluated a logistic regression model and six ML algorithms to predict whether patients will develop intraoperative hypothermia during non-cardiac surgery. We found that both discrimination and calibration varied considerably between logistic regression and the ML algorithms analyzed, rendering only some techniques, such as XGBoost, RF, and logistic regression, suitable for further evaluation in clinical practice. We defined hypothermia using two approaches: first, as hypothermia burden (degrees below 37 °C·h); and second, as point incidence of body temperature below three thresholds. In general, predicting the extremes of hypothermia burden (i.e., whether a patient will likely experience no hypothermia or severe hypothermia) proved to be superior compared to predicting mild/moderate hypothermia and relying on point incidences. Across all tested combinations, XGBoost had superior discrimination and had good calibration.

It has previously been shown that hypothermia, defined as minimum body temperature below one of three thresholds, is a common occurrence during general anesthesia [21]. Although thermal care interventions, such as active prewarming prior to induction of anesthesia and intraoperative forced air warming, are recommend for routine use [11], they are often not performed, either due to equipment shortage, understaffing, or presumed high costs [28]. The occurrence of intraoperative hypothermia is a complex phenomenon attributable to a variety of patient-specific, surgical, and system-specific risk factors, and interactions between these factors are likely [29]. Predicting the occurrence of hypothermia could be useful for targeted resource allocation and could contribute to economic savings attributed to thermal care interventions [4]. In addition, a predictive model able to identify patients at high risk of intraoperative hypothermia could be used to select patients for prospective trials.

While normal core body temperature is about 37 °C [2], temperatures as low as 36 °C are considered normal in anesthetized patients [1]. Recently, it has been demonstrated that intraoperative body temperatures even as low as 35.5 °C were not associated with adverse outcomes [13]. We recently showed that hypothermia defined by body temperature falling below thresholds of 36 °C and 35.5 °C at least at a single time point can be predicted with acceptable performance using logistic regression [14]. However, the definition of hypothermia used in this study may be imprecise when applied to our dataset, as nearly half of all patients in our cohort crossed the <36 °C threshold, even though average body temperature during anesthesia was above >36°. It is biologically plausible that the detrimental effects of hypothermia are related to both the absolute fall of body temperature as well as the duration of exposure. Hypothermia burden, the area under the body temperature below 37 °C curve over time, relates those features and may be clinically more important than hypothermia at single time points. Indeed, it was shown previously that increasing hypothermia burden is associated with higher odds for perioperative blood product transfusion as well as increased hospital length of stay [21].

In our previous study, we predicted hypothermia using only logistic regression, a statistical method used to analyze the association between data points and binary outcomes. ML, however, is an umbrella term for a heterogenous group of computational algorithms that can be used to predict binary or multilabel outcomes, such as the hypothermia burden group. Advantages commonly attributed to ML include the ability to automatically process a large amount of data and fit models with many predictors [18] while imposing fewer restrictions on the dataset. For instance, co-linearities between predictors are automatically uncovered by some ML techniques (e.g., XGBoost, RF, and MLP) but have to be explicitly specified when logistic regression is used [18,30]. In our dataset, those algorithms that were able to automatically detect non-linear relationships outperformed logistic regression. While MLP showed marginal improvements in F1 score (<1%) over logistic regression, the decision tree ensemble methods XGBoost and RF performed considerably better (11% and 5%, respectively). Interestingly, the outcome definition seems to be of particular importance to obtain good model discrimination, as the discrimination of hypothermia defined via threshold temperatures was considerably worse than the prediction of the extreme hypothermia burden quarters (no hypothermia and severe hypothermia), which were more easily detected as those in-between. In addition, some models (MLP and logistic regression) used to predict hypothermia below the three thresholds were severely miscalibrated, possibly related to the relatively small incidence of hypothermia below 35 °C and 35.5 °C. Regarding the prediction of hypothermia burden, XGBoost and RF had acceptable calibration errors, but logistic regression was the best calibrated algorithm. GNB and KNN performed considerably worse in terms of both discrimination and calibration. A possible explanation for the subpar performance of GNB could be that it assumes the independence of variables in the feature set.

Despite the advantages attributed to ML, there is still an open debate on whether ML is superior to traditional logistic regression-based methods for clinical data. It has been claimed that ML methods show no performance benefit over logistic regression [31] and that ML models often lack transparent reporting [32]. In particular, the lack of calibration assessment of ML models has often been criticized [33]. Our data indicate that discrimination can be improved when ML, particularly XGBoost, is used.

### Limitations

Due to the retrospective nature of this study, we had no information on data not routinely recorded at our institution, such as patient core body temperature prior to induction of anesthesia or type and location of the temperature probe. Most anesthesia cases analyzed involved general anesthesia. Only a minuscule proportion of cases had neuraxial or regional anesthesia, which is related to the lack of routine temperature measurements in those cases, and which leads to selection bias in our study. As we only used data from our center, we cannot report on the external validity of our models. In addition, as we primarily intended to compare the predictive performance of ML models to regression analysis using an existing dataset, not all predictor variables are available before surgery. For instance, the use of neuromuscular blockade or vasoactive medication is sometimes not planned for. This will reduce predictive performance if our models are used to predict hypothermia ahead of surgery based on a presumed course of anesthesia. Therefore, the results of our study are not readily translatable into clinical practice and should be externally validated in further studies.

In summary, we built several ML models to predict hypothermia of varying severity and compared their performance to logistic regression. We found better discrimination than logistic regression and acceptable calibration with some models (XGBoost, RF), while others (KNN, GNB) performed significantly worse. Future studies on predicting intraoperative hypothermia using ML should focus on tree-based ML algorithms and validate our findings using a prospective study design.

## Figures and Tables

**Figure 1 jcm-12-04434-f001:**
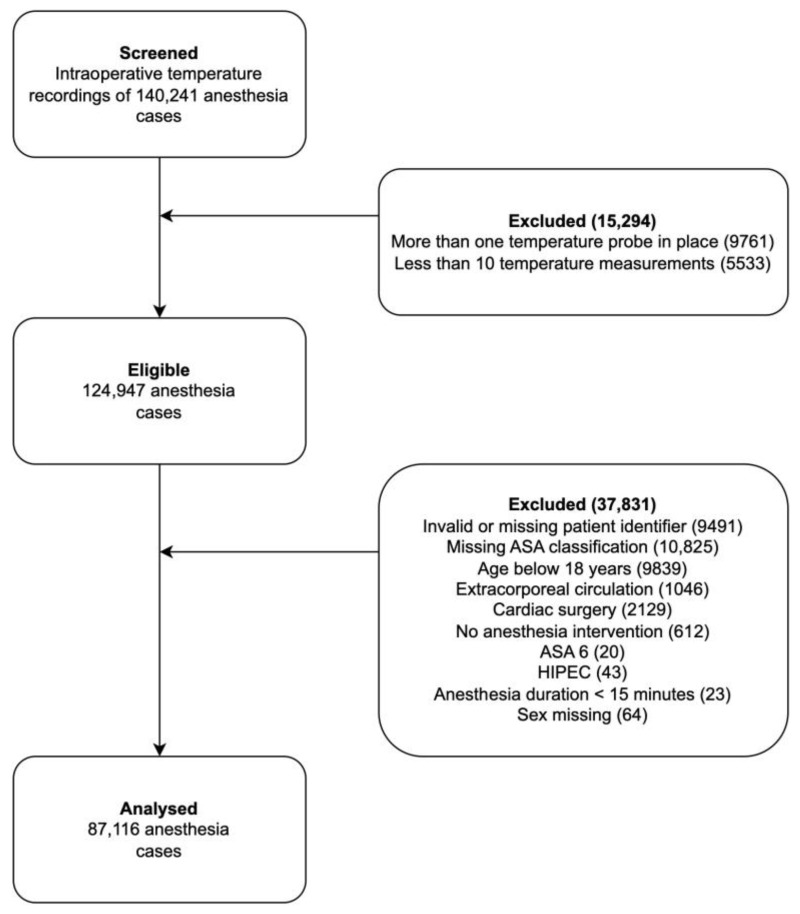
Patient inclusion flow chart. ASA: American Society of Anesthesiologists. HIPEC: Hyperthermic intraperitoneal chemotherapy.

**Figure 2 jcm-12-04434-f002:**
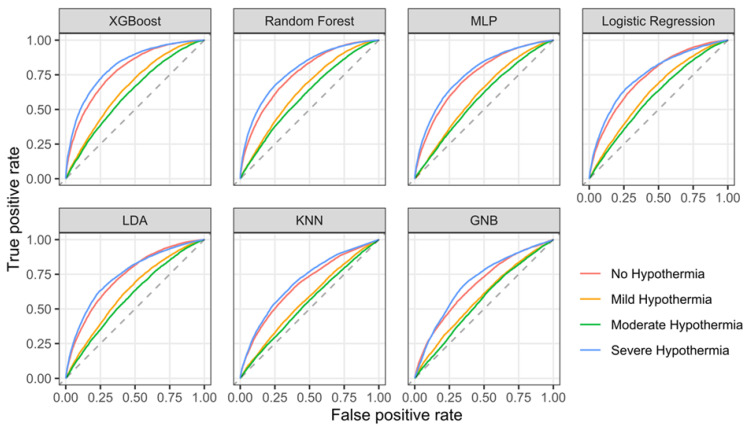
Receiver operating characteristic curve for the prediction of hypothermia burden quartile. MLP: multi-layer perceptron neural network, LDA: linear discriminant analysis, KNN: K-nearest neighbor, GNB: Gaussian naïve Bayes.

**Figure 3 jcm-12-04434-f003:**
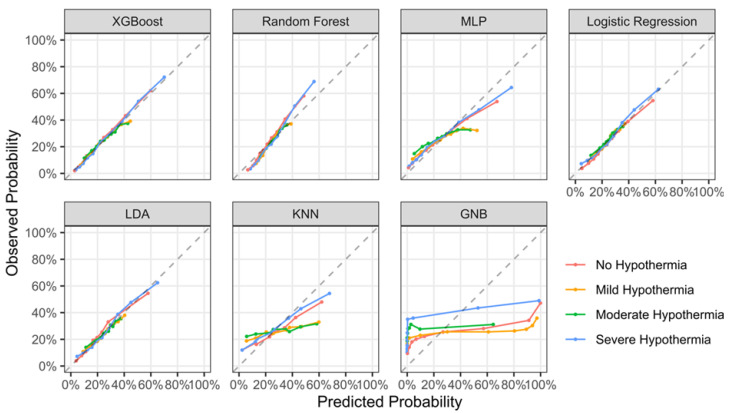
Calibration plots for prediction of hypothermia burden. MLP: multi-layer perceptron neural network; LDA: linear discriminant analysis; KNN: K-nearest neighbor; GNB: Gaussian naïve Bayes.

**Table 1 jcm-12-04434-t001:** Baseline characteristics of study participants.

	No Hypothermia n = 19,484	Mild Hypothermia n = 22,220	Moderate Hypothermia n = 22,800	Severe Hypothermia n = 22,612
**Age (years)**	52 (37–66)	53 (39–67)	55 (41–68)	58 (45–70)
**Male sex**	9076 (47%)	10,547 (47%)	11,520 (51%)	11,785 (52%)
**Weight (kg)**	77 (65–90)	75 (64–88)	75 (65–87)	74 (63–85)
Unknown	2372	2484	2063	1342
**Height (cm)**	170 (164–178)	170 (164–178)	170 (164–178)	170 (164–178)
Unknown	9961	10,791	10,646	9838
**ASA score**				
I	4702 (24%)	5885 (26%)	5509 (24%)	3947 (17%)
II	7788 (40%)	10,110 (45%)	10,396 (46%)	9688 (43%)
III	5856 (30%)	5528 (25%)	6045 (27%)	7709 (34%)
IV	883 (4.5%)	541 (2.4%)	615 (2.7%)	839 (3.7%)
V	255 (1.3%)	156 (0.7%)	235 (1.0%)	429 (1.9%)
**Surgical urgency**				
Elective	11,285 (68%)	14,689 (81%)	15,296 (83%)	14,962 (81%)
Urgent	4071 (25%)	2741 (15%)	2439 (13%)	2335 (13%)
Emergency	1121 (6.8%)	691 (3.8%)	745 (4.0%)	1076 (5.9%)
Unknown	3007	4099	4320	4239
**van Walraven comorbidity score**	0.0 (0.0–4.0)	0.0 (0.0–4.0)	0.0 (0.0–4.0)	0.0 (0.0–4.0)
Unknown	296	377	380	399
**Surgical discipline**				
Oral and maxillofacial surgery	1002 (5.1%)	1131 (5.1%)	1303 (5.7%)	997 (4.4%)
Plastic surgery	903 (4.6%)	937 (4.2%)	1052 (4.6%)	1172 (5.2%)
Head and neck surgery	2320 (12%)	1945 (8.8%)	1465 (6.4%)	661 (2.9%)
Dermatology	136 (0.7%)	269 (1.2%)	239 (1.0%)	100 (0.4%)
Orthopedic and/or trauma surgery	2870 (15%)	4503 (20%)	5789 (25%)	5697 (25%)
Ophthalmology	1266 (6.5%)	1667 (7.5%)	941 (4.1%)	230 (1.0%)
Urology	1504 (7.7%)	1747 (7.9%)	1551 (6.8%)	1610 (7.1%)
General surgery	5822 (30%)	5519 (25%)	5220 (23%)	5235 (23%)
Gynecology	1888 (9.7%)	2369 (11%)	1915 (8.4%)	1440 (6.4%)
Obstetrics	72 (0.4%)	61 (0.3%)	39 (0.2%)	33 (0.1%)
Vascular surgery	613 (3.1%)	512 (2.3%)	706 (3.1%)	1377 (6.1%)
Thoracic surgery	409 (2.1%)	595 (2.7%)	942 (4.1%)	1548 (6.8%)
Neurosurgery	834 (4.3%)	1143 (5.1%)	1888 (8.3%)	2930 (13%)
**Ambient room** **temperature (°C)**	19.99 (19.02–20.95)	19.98 (19.00–20.87)	19.90 (18.99–20.44)	19.18 (18.99–20.05)
Unknown	3008	3560	4546	5793
**Operating room time (min)**	83 (61–121)	98 (76–136)	135 (106–182)	222 (166–310)

All data are given as absolute and relative frequencies or the median and interquartile range. ASA: American Society of Anesthesiologists, kg: kilogram, cm: centimeter.

**Table 2 jcm-12-04434-t002:** Baseline characteristics of study participants. Comparison of model discrimination as measured by area and the receiver operating characteristics curve (AUROC) to predict hypothermia burden. The relative change with respect to the AUROC of the corresponding logistic regression model is given in parentheses.

	Weighted F1 Score	AUROC			
		*No* *Hypothermia*	*Mild* *Hypothermia*	*Moderate* *Hypothermia*	*Severe* *Hypothermia*
**XGBoost**	0.44 (10.74%)	0.781 (6.22%)	0.655 (4.49%)	0.617 (3.75%)	0.812 (8.44%)
**Random Forest**	0.418 (5.15%)	0.756 (2.85%)	0.641 (2.36%)	0.604 (1.55%)	0.784 (4.59%)
**LDA**	0.406 (2.16%)	0.735 (−0.05%)	0.626 (−0.06%)	0.592 (−0.39%)	0.748 (−0.2%)
**MLP**	0.4 (0.58%)	0.738 (0.45%)	0.607 (−3.11%)	0.582 (−2.06%)	0.761 (1.6%)
**Logistic** **Regression**	0.397 (0.00%)	0.735 (0%)	0.627 (0%)	0.594 (0%)	0.749 (0%)
**KNN**	0.362 (−8.97%)	0.676 (−7.98%)	0.568 (−9.38%)	0.542 (−8.8%)	0.699 (−6.7%)
**GNB**	0.32 (−19.50%)	0.673 (−8.48%)	0.58 (−7.45%)	0.558 (−6.19%)	0.694 (−7.34%)

XGBoost: extreme gradient boosting, MLP: multi-layer perceptron neural network, LDA: linear discriminant analysis, KNN: k-nearest neighbor, GNB: Gaussian naïve Bayes.

**Table 3 jcm-12-04434-t003:** Comparison of model calibration to predict hypothermia burden quarter.

	No Hypothermia			Mild Hypothermia			Moderate Hypothermia			Severe Hypothermia		
	Mean	Intercept	Slope	Mean	Intercept	Slope	Mean	Intercept	Slope	Mean	Intercept	Slope
**XGBoost**	0.044	0.118	1.064	−0.033	−0.093	0.939	−0.012	−0.184	0.824	0.005	0.090	1.102
**Random Forest**	0.044	0.521	1.423	−0.037	0.269	1.307	−0.017	0.242	1.260	0.014	0.483	1.510
**MLP**	0.024	−0.331	0.640	−0.093	−0.621	0.429	0.058	−0.591	0.379	0.013	−0.276	0.656
**LDA**	0.061	−0.021	0.920	−0.041	−0.148	0.894	−0.021	−0.250	0.768	0.006	−0.051	0.937
**Logistic Regression**	−0.094	−0.094	1.015	0.011	0.017	1.006	0.049	−0.094	0.864	0.030	0.015	0.978
**KNN**	0.058	−0.846	0.131	−0.124	−0.964	0.071	−0.058	−0.971	0.038	0.135	−0.644	0.152
**GNB**	−0.366	−1.135	0.020	−0.946	−1.059	0.006	1.395	−0.974	0.010	0.635	−0.627	0.062

XGBoost: extreme gradient boosting, MLP: multi-layer perceptron neural network, LDA: linear discriminant analysis, KNN: k-nearest neighbor, GNB: Gaussian naïve Bayes.

## Data Availability

Data are available from the corresponding author upon reasonable request.

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
