# Peer review of "Predicting Intraoperative Hypothermia Burden during Non-Cardiac Surgery: A Retrospective Study Comparing Regression to Six Machine Learning Algorithms"

_jcm, 2023, doi:10.3390/jcm12134434_

Round 1

Reviewer 1 Report

This is an interesting attempt to compare two methods for predicting intraoperative hypothermia.

Please see the notes attached for more details.

Author Response

This article represents a nice try by the authors to compare regression analysis to enhanced machine learning algorithms in predicting intraoperative hypothermia during non- cardiac surgery. They used a tertiary academic medical center’s data base and designed a ML algorithm to predict the patients that are at higher risk to develop intraoperative hypothermia and hence would benefit the most by the implementation of hypothermia preventing strategies, in an environment with resources shortage. Intraoperative hypothermia is an important issue in perioperative care, that can affect a patient’s morbidity. There are specific recommendations on how to minimize its risk that should be followed. In the article there should be no implications that these recommendations can be ignored.

We thank reviewer #1 for their thorough evaluation of our manuscript. We provide per-point answers to the issues raised by them:

The definition for inadvertent intraoperative hypothermia is core body temperature below 36 °C. Why did you choose to define Hypothermia burden as area below the body temperature under 37 °C curve over time?

We agree with reviewer #1 that intraoperative hypothermia is commonly defined as core body temperature below 36 °C at a single time point during surgery. We decided to use body temperature below 37 °C on the temperature curve over time (“hypothermia burden”) as our primary outcome, as this measurement is more clinically meaningful, since it combines not only extent/severity of hypothermia but also duration of hypothermia, and has also been used previously (Sun et al. Intraoperative Core Temperature Patterns, Transfusion Requirement, and Hospital Duration in Patients Warmed with Forced Air. Anesthesiology 2015, doi:10.1097/ALN.0000000000000551). It has been previously shown that a large proportion of surgical patients develops hypothermia rapidly after induction of anesthesia. Of those patients, many remain hypothermic only for a short period time, after which body temperature normalizes as thermal care interventions, such as forced air warming, are put in place. Of those patients, many would be labelled as hypothermic when using the “body temperature below 36 °C for at least one perioperative measurement” as definition, but at the same time “hypothermia burden” as defined by us in our manuscript would be low. However, short term hypothermia such as in those cases does not carry the same clinical significance as longer and/or more severe periods of hypothermia.

Would it have been more appropriate to include only data from patients with body core measurements?

At our institution, intraoperative body temperature is most commonly monitored using esophageal temperature probes placed orally or nasally. Alternatively, urinary catheters with integrated temperature probes are used. Both esophageal as well as bladder temperature closely resemble body core temperature. We thus believe that the majority of temperature measurements in this study appropriately reflect body core temperature. However, information on the individual location of the temperature probe used during anesthesia was not available in our data source.

Would it have been more appropriate at least at this first stage to exclude all the regional anesthesia patients and include only patients under general anesthesia?

We agree with reviewer #1 that the results of our study cannot be translated to patients receiving regional/neuraxial anesthesia, as the number of those patients included in our study was rather low. This reflects the fact that – at our institution – intraoperative body temperature is not routinely monitored in patients with neuraxial anesthesia and almost never in patients with peripheral nerve blocks. However, inclusion and exclusion criteria were predefined for our study and as such, we also included patients with regional anesthesia in our study. Furthermore evidence suggests a comparable incidence of hypothermia in patients with spinal or epidural anesthesia.

Do you believe that a power analysis would have a significant impact on how you interpret your results?

We did not perform a power analysis for our study as we were able to include all patients available in our database. Due to the large number of patients included in our study, we consider it highly unlikely that our study is underpowered. As such, we do not believe that a formal power analysis would have changed the interpretation of our findings.

Do you have Ethics Committee approval for this study (if necessary, in your country), and since you use patients’ data, have they signed informed consent beforehand that their data may be used for such studies in the future? Please state it clearly in the text.

This study was approved by the Ethics Committee the Medical University of Vienna on May 11th, 2021 (protocol code 1402/2021) prior to data export and analysis. The Ethics Committee waived the need to obtain informed consent from included patients due to the retrospective study design. We describe this on pages 10-11, lines 384-387 of our manuscript.

In line 98 you mention 14 variables for types of surgery, but you present 3. Which are the other variables?

We thank reviewer #1 for pointing out the missing information on surgical variables. The missing variables refer to intraoperative positioning. There are 12 binary variables encoding information on intraoperative positioning: supine, prone, trendelenburg, anti-trendelenburg, beachchair, leg extension, lloyd-davis, lithotomy, side, flexed side, other, not documented (all yes/no). We corrected this in our manuscript on page 3, lines 104-109.

In line 99 you mention 9 variables for the type of anesthesia, but you present 8.

We thank reviewer #1 for pointing out our mistake regarding the description of variables for “type of anesthesia”. There are two missing variables, both referring to the location of the peripheral nerve block. The correct list of variables on type of anesthesia can be found on page 3, lines 108-109.

In line 108 can you inform us which are the 24 variables for vital signs that you used?

For each patient, we exported longitudinal vital sign recordings with for systolic, diastolic and mean blood pressure as well as peripheral transcutaneous oxygen saturation, heart rate and pulse frequency (i.e., 6 variables). From this data, we than extracted the vital signs at four time points: 1) the initial vital signs (ie., the first vitals signs measured after the start of the electronic anesthesia protocol), 2) the minimum and 3) maximum vital parameters (up until the documented start of surgery) and 4) the vital signs after induction of anesthesia. As such, we obtained a total of 6 x 4 = 24 variables. The original variable list did erroneously not include “heart rate”. We apologize for this mistake and corrected it on page 3, line 118.

Reviewer 2 Report

Dear Authors,

I would like to congratulate you for this very interesting work and manuscript. After reviewing the manuscript I would like to suggest that some terms regarding AI and machine learning should be written in a more simple language in order to be more clear in readers that are not familiar with AI. Overall, this is a well-written manuscript of a very well organized study that brings AI inside the OR in an attempt to prevent hypothermia which can lead to increased morbidity intraoperatively.

Author Response

I would like to congratulate you for this very interesting work and manuscript. After reviewing the manuscript I would like to suggest that some terms regarding AI and machine learning should be written in a more simple language in order to be more clear in readers that are not familiar with AI. Overall, this is a well-written manuscript of a very well organized study that brings AI inside the OR in an attempt to prevent hypothermia which can lead to increased morbidity intraoperatively.

We thank reviewer #2 for their work and their encouraging statement. We adapted our introduction to machine leaning (page 2, lines 57-62).